# Taste Sensitivity of Elderly People Is Associated with Quality of Life and Inadequate Dietary Intake

**DOI:** 10.3390/nu13051693

**Published:** 2021-05-17

**Authors:** Soyeon Jeon, Yeonhee Kim, Sohyun Min, Mina Song, Sungtaek Son, Seungmin Lee

**Affiliations:** 1Clinical Nutrition Program, Graduate School of Human Environmental Sciences, Yonsei University, Seoul 03722, Korea; kes1647@hanmail.net; 2Department of Food and Nutrition, BK21 FOUR Project, College of Human Ecology, Yonsei University, Seoul 03722, Korea; yh0227@yonsei.ac.kr (Y.K.); minsh@yonsei.ac.kr (S.M.); msyu@yonsei.ac.kr (M.S.); 3Department of Applied Statistics, College of Economics and Commerce, Yonsei University, Seoul 03722, Korea; stson1991@gmail.com

**Keywords:** elderly people, taste recognition, salty taste, sour taste, dietary reference intake, iron, thiamin, quality of life

## Abstract

Aging has been implicated in the alteration of taste acuity. Diet can affect taste sensitivity. We aimed to investigate the types of tastes altered in elderly Korean people and factors associated with taste alteration in relation to dietary intake and other factors. Elderly participants (≥65 years) and young adults were assessed to determine their recognition thresholds (RT) for sweet, salty, bitter, sour, and umami tastes. Elderly participants were further surveyed for dietary intake and non-nutritional factors. Five taste RTs were correlated with age, but only four taste RTs, except sweetness, differed between the elderly participants and young adults. Inadequate intake of iron, thiamin, folic acid, zinc, and phosphorus among the elderly participants was related to elevated taste RT levels, except for bitter taste. In both correlation and regression analyses, only salty and sour RTs were associated with energy, iron, thiamin, fiber, vitamin C, and riboflavin levels in the elderly participants. The elderly participants’ taste RTs exhibited strong associations with quality of life (QOL) but showed partial relationships with physical activity, number of medicine intakes, social gatherings, and education. Taste sensitivity may decrease with age, which is further influenced by insufficient dietary intake, especially iron and thiamin, and QOL.

## 1. Introduction

According to the United States Census Bureau, the number of people aged 65 years and older is expected to increase threefold by 2050. Advances in technology leading to easy access to safe water and disease prevention coverage and a stabilized socioeconomic status, such as wide coverage of legal health insurance, effective health policy, and health education, are considered to increase life expectancy [1]. Along with this global trend, there has been an increasing need to maintain good health and lower the risk of developing diseases in elderly people [2]. Deterioration of oral health, altered taste ability, and lack of competencies in socioeconomic and psychological factors have been observed in elderly people [3]. Loss of appetite and food intake can lead to malnutrition in elderly people [4]. Taste alteration may contribute to reduced appetite and food intake in older people [5].

Aging has been implicated in the reduction of taste acuity. The sensation of taste gradually decreased around the age of 60 years [6]. The size of taste buds and the number of taste cells were found to be decreased in older mice [7]. Changes in taste acuity in elderly people could be, in part, due to the loss of taste receptors by aging-related physiology [8]. The increase in the use of medication in elderly people may also affect the oral conditions influencing the sense of taste [9]. Certain drugs, especially when multiple drugs are used for treatment, trigger taste alteration in elderly people [10]. The prevalence of drug-related taste alterations is as high as 11% [11]. Although the exact mechanism of the effect of aging on taste sensation is not well understood, aging appears to clearly influence taste sensation. 

Besides aging, multiple genetic and environmental factors have been associated with taste alteration. Heavy consumption of alcoholic beverages is a factor that lowers the sensitivity to sweet taste, causing a higher intake of sugar [12]. Poor oral conditions such as cavity infections, damage to the central or peripheral nerves, and decreased amounts of saliva have also been proposed to cause taste alterations [13]. Malnutrition can be the cause and result of alterations in taste perception [14]. Moreover, zinc has significant effects on decreased taste sensitivity [15]. 

In addition to these considerations, non-nutritional factors have been proposed to cause taste alterations. Quality of life (QOL) has been used as an assessment tool for individuals’ physical, psychological, and environmental circumstances [16]. Multisensory dysfunctions were associated with lower QOL and depression in the British elderly people aged 52 years and older [17]. In addition, QOL scores and nutritional conditions were suggested to affect the food choices of elderly people [18]. As only a few studies have investigated the relationships between QOL and taste changes in elderly people, further research is warranted to evaluate these associations further. 

Taste dysfunction has rarely been investigated in Asian populations [19]. Significant differences have been observed in taste recognition thresholds between different ethnic groups in the same region [20]. A study by Yang et al. showed that increasing age had no effect on impaired taste function in 90 healthy Chinese individuals, suggesting that lifestyle factors such as dietary habits may influence taste changes [21]. However, this study was conducted only in individuals aged younger than 65 years. Recent studies reported that age and sex are associated with taste alterations in 240 healthy Taiwanese individuals [19]. However, in this study, none of the individuals’ eating habits and nutritional intake were considered, and only four basic tastes (sweet, salty, bitter, and sour) were measured. Therefore, the relationship between taste perception and dietary nutrient intake requires further investigation. Accordingly, it is necessary to investigate the dietary styles and food intake of the study participants to identify the significant factors that affect taste changes in the elderly population.

In summary, taste alteration can cause loss of appetite and serious health problems in elderly people. To this end, we aimed to assess the recognition of five basic tastes in the elderly population compared with that in young adults and to investigate the relationship between the alteration of basic tastes and factors such as the status of dietary nutrient intake and lifestyle in these older participants, which would ultimately aid in a deeper understanding of age-related taste changes and dietary recommendations in elderly people experiencing taste alteration to maintain and/or enhance their nutritional status and to prevent and/or alleviate age-associated malnutrition. 

## 2. Materials and Methods

### 2.1. Study Participants

Ninety-seven older individuals aged 65 years and older were recruited from a senior welfare center in Seoul, Korea, between October and November 2015. Forty-nine young adults aged 20–29 years were recruited from Yonsei University, Seoul, Korea. This study was approved by the Institutional Review Board (IRB) of the Human Research Protection Center at Yonsei University in Seoul, Korea (approval number: 1040917-201507-SB-172-04). Twenty-nine older individuals were excluded owing to missing data (*n* = 10) and taste threshold non-responders or response errors (*n* = 19). Twenty-two healthy young adults in their twenties were additionally recruited as approved by the IRB (approval number: 7001988-201708-HR-229-02), and none were excluded. The participants were recruited on a voluntary basis in response to notices on bulletin boards. The protocol complied with the Declaration of Helsinki for Medical Research involving human subjects. Participants were considered eligible if they had adequate cognition to complete the interviews. Participants diagnosed with dental diseases and dementia, which included Alzheimer’s disease and vascular dementia, were excluded. Written informed consent was obtained from all study participants. Accordingly, 168 participants, including 97 elderly people and 71 adults, were recruited; however, a total of 139 individuals, including 68 elderly people and 71 younger adults, were included in the final analysis.

### 2.2. Data Collection

#### 2.2.1. General Characteristics and Health-Related Behaviors Questionnaire 

A questionnaire including general characteristics such as age, sex, height (cm), weight (kg), and health-related behaviors including exercise, alcohol consumption, and smoking data with a 3-point scale rating was conducted with both the young adult and elderly group. Exercise was categorized based on the frequency of exercise per week: none, once to twice per week, and more than three times per week. Alcohol consumption was categorized as “no alcohol intake”, “less than one glass per month in the past year”, and “more than one glass per month”, and smoking status was reported as “never smoked”, “former smoker” and “currently smokes”. Participants’ physical activity levels were assessed using the International Physical Activity Questionnaire [22]. The estimated energy requirement (EER) was calculated using energy intake, energy expenditure, age, sex, weight, height, and physical activity level. The equations for adults aged 19 years and older were as follows: EER (kcal/day) = 662 − (9.53 × age (y)) + PA × ((15.91 × weight (kg)) + (539.6 × height (m))) for men, and EER (kcal/day) = 354 − (6.91 × age (y)) + PA × ((9.36 × weight (kg)) + (726 × height (m))) for women, where PA is the physical activity coefficient [23].

#### 2.2.2. QOL Assessment

The QOL assessment was only conducted with the elderly group. The objective QOL factors such as marital status and living situation including self-rated standard of living, number of medicine intakes, cohabitation (living with at least one person), and social participation, including hobbies, social gatherings, voluntary service, and education (senior college/lifelong education/senior center participation) were assessed using the Manchester Assessment of Quality of Life through interviews [24]. Social participation was evaluated based on five choices of “barely any”, “once or twice a year”, “once or twice a month”, “once or twice a week”, and “almost every day”. The subjective QOL was assessed using the Korean WHOQOL-BREF, a modified version of the World Health Organization World Health Quality of Life Assessment Instrument-100 (WHOQOL-100) [25]. The Korean WHOQOL-BREF is a self-report test that assesses an individual’s perception of their subjective QOL in the past two weeks. This test consists of four domains: physical health (8 questions), psychological health (6 questions), social relationships (2 questions), and environmental aspects (8 questions), with a total of 24 facets. Each question was rated on a 5-point scale (1 = negative extreme and 5 = positive extreme), and the participants were asked to choose the most accurate choice. The scale scores were as follows: “strongly disagree” = 1, “disagree” = 2, “mixed” = 3, “agree” = 4, and “strongly agree” = 5; in case of negatively worded questions with reversed scores, the chosen score subtracted from 6 was used instead. The scores for each subdomain were assessed by multiplying the mean score of each subdomain by 4; thus, the scores for each subdomain ranged from 4 to 20. The total score was assessed by adding the four scores for each subdomain. For analysis, the participants were divided into three groups according to the quartiles of the WHOQOL-BREF score distribution. The Cronbach’s α value of the conducted test was 0.855, while those of the four subdomains were 0.724 for physical health, 0.855 for psychological health, 0.370 for social relationships, and 0.826 for environmental aspects.

#### 2.2.3. Food Intake and Nutrient Intake Questionnaire

Food intake frequency was assessed in the elderly group using the interviewer-administered food frequency questionnaire developed for the Korean Genome and Epidemiology Study. The questionnaire contained 96 food groups provided by the National Institutes of Health. The participants were asked to determine the frequency of intake of the given food groups over the past year. The choices were “seldom”, “once a month”, “2–3 times a month”, “1–2 times a week”, “3–4 times a week”, “5–6 times a week”, “once a day”, “twice a day”, and “three times a day”. For the foods consumed, the average portion size of the food items was investigated. The average nutrient intake per day for each participant was calculated using the Computer-Aided Nutritional Analysis Program 5.0 (CAN pro 5.0) of the Korean Nutrition Society. The average daily nutrient intake was assessed by multiplying the intake frequency by the intake amount per serving. The 2020 Dietary Reference for Koreans published by the Ministry of Health and Welfare and the Korean Nutrition Society was used to assess the adequacy ratio of nutrient intake, including energy, protein, fiber, vitamin A, vitamin C, thiamin, riboflavin, niacin, vitamin B_6_, folate, vitamin B_12_, calcium, phosphorous, iron, and zinc. 

### 2.3. Taste Threshold Test

The taste threshold test was conducted for the five basic tastes (sweet, salty, bitter, sour, and umami) based on the “whole mouth sip and spit” method, as described by Hazelhof et al. [26]. The solutes were sucrose (CJ Cheiljedang, Seoul, Korea) for sweet, sodium chloride (Hanju, Seoul, Korea) for salty, caffeine (Samchun Chemicals, Seoul, Korea) for bitter, citric acid (Sigma-Aldrich, Darmstadt, Germany) for sour, and monosodium glutamate (MSG) “Miwon” (Daesang, Seoul, Korea) for umami taste. Each compound was dissolved in deionized water to six consecutive dilutions: the solute for sweet taste was diluted at concentrations in the range 5–400 mM, that for salty at 0.9–300 mM, that for bitter at 0.2–26.8 mM, that for sour at 0.005–9.89 mM [27], and that for umami at 0.3–76.3 mM [28]. Two pilot surveys were conducted to determine the ranges of the concentrations of each solute. Prior to testing, the solutions were freshly prepared every week and stored at 4 °C in individual bottles. The participants were provided with one cup of distilled water to rinse their mouth, one empty cup for expectoration, and disposable cups with stimulant solutions in ascending concentrations. The participants rinsed their mouths before each session and trial and then spit into the expectorating cup. The samples were held and swirled around in the mouth to cover the entire tongue and then spit out. The participants indicated which cup tasted different from the blank and which taste quality it was. The recognition threshold (RT) was evaluated as the minimum concentration of the taste at which participants answered the correct taste. The participants who did not respond to the taste stimuli (non-responder) or responded to an incorrect taste (response error) in the taste threshold test were excluded in the statistical analysis. 

### 2.4. Statistical Analysis

All statistical analyses were performed using IBM SPSS Statistics 25 (IBM Corp., New York, USA) and R version 3.4.1 (R Foundation for Statistical Computing, Vienna, Austria). The normality of the data was assessed using the Shapiro‒Wilk test and quantile-quantile (Q-Q) plots. Despite statistically significant results from the Shapiro‒Wilk test, the variables did not show notable deviation from normality in the Q-Q plots. Continuous data were expressed as the mean ± standard deviation, and categorical data were expressed as a percentage. The general characteristics of the young adults and elderly participants were compared using Fisher’s exact test and independent sample Student’s *t*-test. QOL and nutrient and food intake between male and female elderly individuals were compared using the independent sample Student’s *t*-test. The correlation between age and taste threshold was analyzed using Pearson’s correlation coefficient. The relationship between taste thresholds and nutrient and food intake of elderly participants was studied using Spearman’s correlation analysis. The elderly participants were categorized into two groups based on their RT for NaCl. The differences in the daily nutrient intake of these two groups were compared using the independent sample Student’s *t*-test. The significance of the relationship between each factor and each taste threshold was studied using univariate linear regression analysis.

## 3. Results

### 3.1. Comparison of Characteristics between the Young Adult and Elderly Participants 

Table 1 shows the overall features of the participants in our study. No significant differences were observed in sex proportion, body mass index (BMI), and frequency of exercise and smoking between young adults and the elderly participants. Although the mean BMI between young adults and elderly participants was not different, BMI categorization showed that the elderly participants were significantly more overweight or obese than the young adults. The average age and number of medicine intake were significantly higher in the elderly participants than in the young adults. Young adults showed a higher drinking frequency than the elderly participants.

### 3.2. Comparison of Taste Thresholds between the Young Adults and the Elderly Participants

The RTs for salty, bitter, sour, and umami tastes were significantly higher in the elderly participants than in the young adults; sweet RT was not significantly different between the two groups (Figure 1). However, when a correlation analysis was performed between age and the level of RT, sweet taste was also associated with age along with all other basic tastes (Figure 2). When sex differences in tastes in the elderly participants were examined, no significant differences in the RTs for all tested tastes were detected between elderly men and women (Figure 3). Thus, although age was associated with an increase in all taste RTs, salty, bitter, sour, and umami tastes might be strongly affected by age; sweet taste might be less influenced; and sex was not influential in the taste alteration.

### 3.3. Assessment of Dietary Intake in the Elderly Participants

Prior to the analysis of taste RT levels according to the dietary intake status of the elderly participants, dietary intake of energy, protein, fiber, vitamins A and C, thiamin, riboflavin, niacin, vitamin B_6_, folic acid, vitamin B_12_, calcium, phosphorus, sodium, iron, and zinc were assessed and compared with the respective Korean dietary reference intake of each elderly participant (Appendix A). The elderly participants were grouped based on their energy or nutrient intake levels. Taste RTs were compared among the groups that had inadequate, adequate, or excessive intake levels. The inadequate dietary intake of thiamin, folic acid, vitamin B_12_, phosphorus, sodium, iron, and zinc increased one or more RT levels of taste, except for bitter taste (Table 2). Bitter taste RT did not appear to differ according to dietary intake level. Except for bitter taste, all other tastes were less recognized by the elderly participants with inadequate dietary intake of iron. In addition, the inadequate dietary intake of some nutrients also elevated part of the tastes’ RTs: folic acid or phosphorus for sweet and sour, zinc for sweet and salty, and thiamin for sour (Table 2). Vitamin B_12_ for salty and sodium for sour showed marginal differences in their respective RT levels. Intake levels of the rest of the dietary nutrients, including energy, protein, fiber, vitamin A, vitamin C, riboflavin, niacin, and calcium, did not differ between the two groups (Table 2). These results suggest that the sensitivities of sweet, salty, sour, and umami were strongly influenced by inadequate iron intake and partly by other nutrient deficiencies, including zinc, folic acid, phosphorus, and thiamin. 

### 3.4. Correlation between Taste Thresholds and Nutrient Intake in the Elderly Participants

To further investigate the linear associations between dietary nutrient intake and taste RT levels, we performed correlation and regression analyses (Table 3). Among the tested RTs, only salty and sour RT levels showed linear associations with dietary intake. Salty RT was negatively associated with dietary intake of iron, thiamin, vitamin C, fiber, and riboflavin in both analyses (Table 3). Sour RT was negatively associated with iron, thiamin, vitamin C, fiber, and energy (Table 3). Taken together, iron, thiamin, vitamin C, and fiber were found to be linearly related to both salty and sour RT, but the others were not. Our findings suggest that the elevation of salty RT and sour RT might be strongly affected by low intake of iron, thiamin, vitamin C, and fiber. Other taste alterations may be less likely to be affected by dietary intake.

### 3.5. Investigation through Non-Nutritional Factors Affecting Taste Thresholds in the Elderly Participants

To assess whether non-nutritional factors were also related to taste alterations, the relationship between other variables, including body weight, BMI, QOL, physical activity, smoking, alcohol consumption, living status, and taste RT levels were tested using correlation and linear regression analyses (Table 4). Negative correlations were observed between salty RT and social gatherings or education, bitter RT and social gatherings, sour RT and social gatherings, and umami RT and physical activity. Positive relationships were found between bitter RT and BMI and the number of medicine intakes. QOL scores were negatively associated with all tested taste thresholds. The reciprocal causality relationships between QOL and taste thresholds were also examined (Appendix A). Our findings implied that several non-nutritional elements might affect some taste RTs of the elderly participants, but QOL scores were investigated to have the relationship with all taste senses of the elderly individuals. 

## 4. Discussion

Many studies have implicated taste alterations in elderly people, but there have been no consistent findings on the types of tastes altered [26,29,30,31,32,33]. According to a recent review, bitter, sour, and umami are the most affected tastes, while the changes to salty and sweet are still questionable [34]. Some researchers have reported significant changes in the sweet, salty, bitter, sour, and umami tastes in elderly people [30,35]. Our results also demonstrated that the recognition of all basic tastes was correlated with age. However, when the average RTs were compared between young adults and the elderly people, no significant difference was observed in the sweet RT despite significant differences in the RTs of salty, bitter, sour, and umami tastes. Hence, sweet taste recognition might be less influenced by age. Similarly, relatively less alteration in sweet taste recognition in elderly people has also been reported in previous studies. For example, Donini et al. reported that, with increasing age, sweet taste was less changed, but saltiness showed the highest increase in sensory impairment among the taste senses [3]. On the other hand, according to Shin et al., increasing ages might be related to stronger reactions to sweet taste, while salty, sour, and bitter tastes are less related to aging [7]. In the study by Barragán et al., taste thresholds for bitter and sour stimuli were more likely to be associated with increased age compared with other taste senses [36]. Mojet et al. reported that elderly men detected sweet and sour stimuli at a stronger intensity than did younger adults [26]. Although the types of tastes affected are inconsistent in these examinations, the decrease in the sensitivity of tastes with increasing age appears to be consistent. 

Dietary nutrient intake can affect taste recognition and possibly the health of elderly people. In our study, the elderly participants with inadequate iron intake showed marked decreases in taste recognition in most of the tested tastes, including sweet, salty, sour, and umami, with the exception of the bitter taste. On the contrary, the bitter taste’s RT was not affected by any tested nutrient inadequacy. This might indicate that adequate dietary iron intake might alleviate and/or prevent alterations in most age-related taste alterations, such as sweet, salty, sour, and umami tastes. Iron is an essential trace element in cell proliferation [37]. In fact, iron deficiency in the diet of elderly people might negatively influence the proliferation of taste cells, considering their relatively rapid turnover rates. Taste receptor cells, located in taste buds, are responsible for identifying the taste compounds and have a turnover rate of 8–12 days [38]. After the damage to the taste buds, regeneration begins within a week, and taste receptor cells are revived in 2 weeks [39]. In a study using electrogustometry, 70% of patients with serum iron deficiency (mean age, 56 years) reported taste disturbance, especially for sour taste; however, supplementation with iron had no effect on elevating taste perceptions. [40]. Moreover, the role of iron has also been implicated in supporting oral health [41]. Iron deficiency is associated with a higher prevalence of oral manifestations, including a burning sensation in the oral mucosa, lingual varicosity, dry mouth, and taste dysfunction [42]. 

Besides iron, a strong negative association of dietary thiamin, vitamin C, and fiber with salty or sour RTs was detected in the elderly participants. Several studies reported possible connections between some of these nutrient deficiencies and taste alteration [43,44]. Thiamin deficiency has been reported to influence appetite loss in elderly patients [43], implying that this association may be related to malnutrition, progressing to worse taste sensitivity [5]. In another study, thiamin deficiency was suggested to cause loss of neurons and dendritic spines in the hippocampus [45], possibly diminishing neural interactions [46], causing taste alteration in elderly people. Moreover, thiamin deficiency may lead to worse oral status by causing dysfunctions of the oral mucosa, tongue, and teeth dentine [47]. Langan et al. reported that elderly people with enhanced oral sanitation showed higher taste sensitivity and dietary intake of thiamin than the control group [48], proposing the possible performance of thiamin in oral health and higher taste sensitivity. Fibers as prebiotics may influence taste alteration, partly by maintaining healthy gut microbiota. Previously, the gastrointestinal microbiota were suggested to influence the eating behavior of hosts and affect food preferences [49]. When gut health was not maintained in patients with bowel inflammation, salty taste sensitivity declined [44]. Vitamin C has been suggested to influence the adrenal cortex [50], and adrenal cortical insufficiency is one of the factors that elevates salty taste sensitivity [51]. Deteriorated dental health status of the elderly population has been suggested to decrease nutritional consumption from diets, and declined dietary intake of vitamin C was exhibited in the edentate elderly people compared to dentate elderly people. Moreover, decreased dietary intake of vitamin C and serum level may induce higher prevalence of periodontal diseases in elderly people [52], which may imply the possible role of vitamin C in oral health. As oral diseases have been suggested to be the one of the most common causes of taste disorders in elderly people [14], these findings may support the correlation between vitamin C deficiency and lower taste sensitivity. Overall, our study provides evidence that sufficient dietary intake of iron, thiamin, fibers, and vitamin C might alleviate taste alteration in elderly people.

In our study, among the five basic tastes, salty and sour tastes were particularly affected in the elderly people based on their nutrient intake status. The sensation of salty and sour tastes requires Na^+^ or H^+^ ions to be recognized by specific channels and contribute to the production of action potential for taste sensation [53]. Saliva may play an important role in salty and sour taste perception by aiding in the ionization of molecules and/or affecting the taste recognition levels [54]. In addition, increased salivary components such as salivary buffers and salivary flow rate may decrease free H^+^ ions in sour taste detection [54]. A study reported that patients with taste dysfunctions showed a higher salivary flow rate compared to normal people [55], indicating the vital role of saliva in taste recognition. In addition, a reduction in ghrelin with aging might be attributed to the reduced salty and sour taste recognition in elderly people. Shin et al. reported that when the growth hormone secretagogue receptor, which is a ligand-binding subject of ghrelin, was removed in the mouth taste cells, the mice showed decreased sensitivity to salty and sour tastes [56], indicating that ghrelin found in taste buds mediates salty and sour taste perception. Ghrelin serum levels decline with aging [57] and iron deficiency [58]. Insufficient iron intake also induces salivary gland disorders [59]. Furthermore, mice fed an iron-deficient diet showed a higher intake of sodium than those fed with other minerals [60]. Therefore, monitoring salty taste thresholds and dietary intervention with iron rich food in elderly people might help alleviate salty taste alteration and reduce high salt intake and related health concerns. 

In addition, we showed that BMI, medication intake, and physical activity were related to taste of elderly people and the QOL score was correlated with all taste RTs. Consistent with our study, the use of medications, the presence of diseases, and physical and socio-psychological changes have been suggested to affect the sensation of taste in elderly people [14]. Aging has been proposed to be related to lower QOL scores due to increased disease risks and social failures [61]. Lower QOL and more depressive signs of older British people were associated with gustatory changes [17]. In addition, medications commonly used in elderly people such as cardiovascular, corticosteroids, psychotropic agents, antibacterial agents, and metabolic agents have been suggested to cause taste alteration [14]. Drugs may affect the taste perception of elderly people through influencing the nervous system such as peripheral receptors or the brain, being secreted into saliva and producing their own tastes, or causing dry mouth syndrome by changing the salivary flow rate and buffers [9,62]. Our results showed that the number of medicines consumed may lower bitter taste sensitivity in the elderly. In contrast, some reported that the number of medications had no connection with taste sensitivity in the elderly population [63,64]. In addition, we observed a negative association between physical activity and umami taste RT in elderly participants. Feeney et al. suggested that men performing routine moderate-to-high-intensity exercises sensed umami taste more strongly than men who did not perform exercise [65]. On the other hand, Horio et al. showed no significant relationship between high-intensity exercises and preference in umami taste in men [66]. To sum up, consideration of the non-nutritional variables, especially QOL, the number of medicine intakes of an individual and physical activity may aid in managing the taste sensitivity of the elderly population. 

Taste dysfunctions such as impaired taste senses or decreased taste sensitivity in the elderly population may lead to appetite loss and nutritional deficiency, so sensory interventions such as flavor enhancement and taste intensification to increase food consumption and QOL have been recommended [67]. In addition, intense tastes have been preferred in the elderly people with taste alterations [68]. Intensification of umami taste increased energy consumption in the elderly patients [67]. Consuming food with high zinc levels has been proven to improve salty taste sensitivity in elderly people [69]. In our data, adequate nutrient intake such as iron and thiamin were suggested to maintain taste recognition of the older people especially in salty and sour tastes. Furthermore, in diet-induced obese rats, supplementing prebiotics was associated with lower preference of sweet taste [70]. 

In our study, elderly participants showed low recognition of salty, bitter, sour, and umami tastes compared with young adults. Insufficient dietary intake of elderly people appeared to alter taste sensitivities, especially salty and sour tastes. Inadequate diet of iron, thiamin, vitamin C, and fibers might worsen the taste sensitivity of elderly people. Our findings may be used as basic data to understand associated factors of taste changes and aid in preventing taste loss in elderly people by providing adequate nutrients and improving quality of life. 

## Figures and Tables

**Figure 1 nutrients-13-01693-f001:**
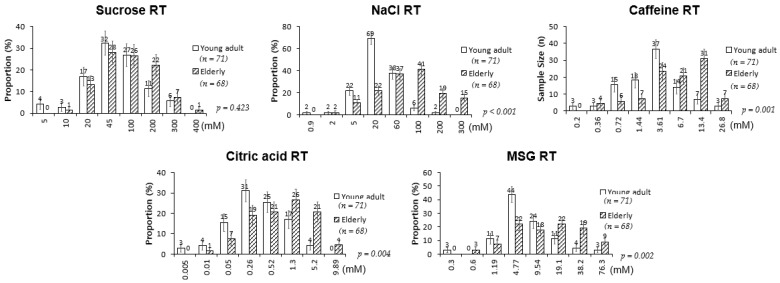
Recognition threshold values of five basic tastes among young adults and elderly participants. *p*-values calculated using Fisher’s exact test (categorical factors). Error bars represent standard errors.

**Figure 2 nutrients-13-01693-f002:**
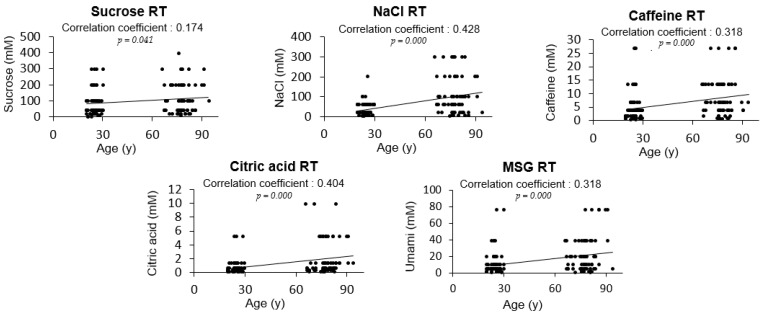
Correlation analysis between age and the five tastes among young adults (*n* = 71) and elderly subjects (*n* = 68). Significant results are shown (*p* < 0.05, Pearson’s correlation analysis).

**Figure 3 nutrients-13-01693-f003:**
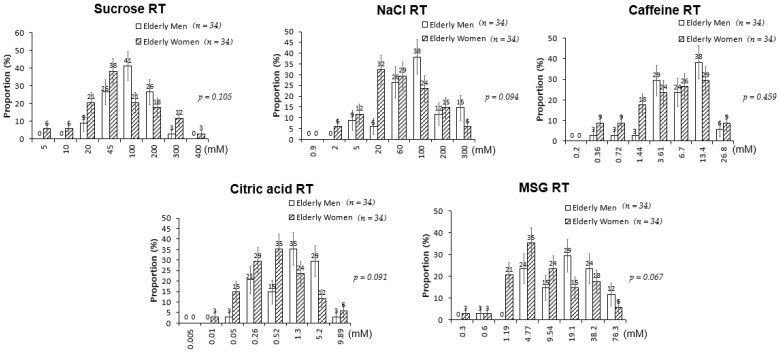
Recognition thresholds for the five tastes in elderly men (*n* = 34) and women (*n* = 34). *p*-values calculated using Fisher’s exact test (categorical factors). Error bars represent standard error.

**Table 1 nutrients-13-01693-t001:** General characteristics of young adults and elderly participants.

Characteristics	Young Adults (*n* = 71)	Elderly (*n* = 68)	*p* ^d^
Sex (Male)	30 (42.3) ^b^	34 (50.0)	0.397 ^e^
Age (y)	24.7 ± 2.5 ^c^	77.9 ± 6.1	0.000
Height (cm)	167.1 ± 7.5	159.2 ± 7.6	0.000
Weight (kg)	59.4 ± 9.8	58.0 ± 9.0	0.000
BMI (kg/m^2^) ^a^	21.2 ± 2.2	22.8 ± 2.8	0.377
Low (<18.5)	6 (8.50)	4 (5.90)	0.000 ^e^
Normal (18.5–22.9)	53 (74.6)	29 (42.6)	
Overweight (23.0–24.9)	9 (12.7)	19 (27.9)	
Obese (≥25)	3 (4.20)	16 (23.5)	
Exercise (more than once a week)	43 (60.6)	48 (70.6)	0.214
Drinking (more than once a month)	66 (93.0)	31 (45.6)	0.000
Smoking (more than one time)	12 (16.9)	20 (29.4)	0.080
Number of medicine intake (≥1)	16 (22.5)	58 (85.3)	0.000

^a^ BMI: body mass index; ^b^
*n* (%); ^c^ mean ± SD; ^d^ significant difference observed between young adults and elderly participants; ^e^ Fisher’s exact test (categorical factors).

**Table 2 nutrients-13-01693-t002:** Comparison of taste thresholds in the elderly participants according to their dietary status (inadequate, adequate, and excessive) for each nutrient.

Parameters		Sucrose RT (mM)	NaCl RT (mM)	Caffeine RT (mM)	Citric Acid RT (mM)	Umami RT (mM)
Energy (% EER)	Inadequate (*n* = 36)	121.9 ± 96.3	110.3 ± 87.4	9.0 ± 6.5	2.3 ± 3.0	22.5 ± 23.1
	Adequate (*n* = 28)	105.5 ± 90.4	101.3 ± 93.4	8.2 ± 7.9	1.7 ± 1.9	19.8 ± 20.1
	Excessive (*n* = 4)	100.0 ± 0.0	60.0 ± 32.7	6.8 ± 5.3	1.9 ± 2.2	16.7 ± 15.8
Protein (% RNI)	Inadequate (*n* = 28)	132.9 ± 100.6	122.1 ± 89.5	9.8 ± 6.5	3.0 ± 3.2	23.3 ± 22.8
	Adequate (*n* = 21)	105.7 ± 95.6	82.4 ± 68.4	7.5 ± 8.0	1.1 ± 1.5	18.9 ± 22.3
	Excessive (*n* = 19)	95.0 ± 64.9	99.8 ± 101.6	7.8 ± 6.6	1.6 ± 1.95	20.2 ± 18.8
Fiber (% AI)	Inadequate (*n* = 31)	128.5 ± 99.6	122.7 ± 87.5	9.3 ± 6.6	2.0 ± 3.12	24.9 ± 24.0
	Adequate (*n = 31)*	119.5 ± 94.9	102.3 ± 100.7	7.6 ± 7.7	1.1 ± 1.4	18.2 ± 22.0
	Excessive (*n = 15)*	75.3 ± 49.2	66.1 ± 53.2	8.2 ± 7.0	1.9 ± 2.1	17.3 ± 12.7
Vitamin A (% RNI)	Inadequate (*n = 51)*	125.4 ± 97.1	108.8 ± 85.6	8.9 ± 7.6	2.1 ± 2.7	21.1 ± 22.4
	Adequate (*n* = 11)	82.7 ± 65.8	95.9 ± 107.9	5.8 ± 4.4	1.5 ± 1.9	20.1 ± 21.3
	Excessive (*n* = 6)	73.3 ± 41.3	73.7 ± 70.8	10.2 ± 5.3	2.3 ± 2.3	23.1 ± 13.0
Vitamin C (% RNI)	Inadequate (*n* = 34)	124.4 ± 96.2	119.6 ± 84.3	8.8 ± 6.5	2.5 ± 3.0	24.3 ± 23.4
	Adequate (*n* = 14)	118.9 ± 108.5	96.1 ± 93.2	6.3 ± 7.3	1.6 ± 2.0	19.1 ± 20.0
	Excessive (*n* = 20)	92.5 ± 64.2	81.9 ± 88.5	9.6 ± 7.6	1.4 ± 1.7	17.1 ± 18.6
Thiamin (% RNI)	Inadequate (*n* = 13)	155.8 ± 113.5	147.7 ± 107.3	10.9 ± 6.2	4.9 ± 3.4^b^	29.8 ± 24.5
	Adequate (*n* = 20)	109.3 ± 80.3	100.3 ± 65.5	8.4 ± 6.5	1.1 ± 1.5^a^	19.9 ± 22.8
	Excessive (*n* = 35)	101.0 ± 84.7	89.2 ± 87.9	7.7 ± 7.5	1.4 ± 1.8^a^	18.5 ± 18.9
Riboflavin (% RNI)	Inadequate (*n* = 32)	125.3 ± 96.5	122.0 ± 85.6	9.5 ± 6.4	2.7 ± 3.1	24.5 ± 23.7
	Adequate (*n* = 18)	121.4 ± 101.8	105.8 ± 95.2	7.9 ± 8.3	0.9 ± 1.2	19.0 ± 23.7
	Excessive (*n* = 18)	86.1 ± 62.3	68.7 ± 77.1	7.3 ± 6.7	2.0 ± 2.1	17.0 ± 13.0
Niacin (% RNI)	Inadequate (*n* = 32)	122.8 ± 98.2	116.4 ± 86.2	9.1 ± 6.6	2.7 ± 3.1	23.4 ± 23.9
	Adequate (*n* = 25)	113.6 ± 94.7	96.4 ± 88.4	8.3 ± 8.2	1.2 ± 1.6	18.2 ± 17.6
	Excessive (*n* = 11)	88.6 ± 51.1	82.9 ± 92.5	7.2 ± 5.5	1.9 ± 2.2	20.9 ± 22.6
Vitamin B_6_ (% RNI)	Inadequate (*n* = 26)	140.6 ± 100.2	130.0 ± 88.8	9.2 ± 5.8	3.0 ± 3.2	22.0 ± 20.8
	Adequate (*n* = 20)	88.3 ± 70.2	77.3 ± 63.7	7.5 ± 8.9	1.3 ± 1.8	19.6 ± 22.6
	Excessive (*n* = 22)	105.7 ± 90.5	96.5 ± 99.5	8.6 ± 6.7	1.5 ± 1.8	21.3 ± 21.8
Folic acid (% RNI)	Inadequate (*n* = 22)	154.1 ± 102.4^b^	135.4 ± 96.6	10.1 ± 5.5	3.2 ± 3.4^b^	24.3 ± 21.8
	Adequate (*n* = 21)	85.0 ± 88.9^a^	74.8 ± 44.0	6.2 ± 8.0	1.1 ± 1.8^a^	18.7 ± 22.4
	Excessive (*n* = 25)	102.8 ± 69.4^ab^	99.9 ± 100.1	9.0 ± 7.1	1.7 ± 1.9^ab^	20.3 ± 20.7
Vitamin B_12_ (% RNI)	Inadequate (*n* = 4)	200.0 ± 81.6	225.0 ± 50.0^b^	5.2 ± 1.8	2.3 ± 2.0	42.9 ± 24.0
	Adequate (*n* = 10)	140.0 ± 121.3	146.0 ± 107^ab^	10.4 ± 4.0	3.9 ± 3.8	22.5 ± 21.5
	Excessive (*n* = 54)	103.0 ± 81.8	86.8 ± 76.5^a^	8.4 ± 7.6	1.7 ± 2.2	19.2 ± 20.7
Calcium (% RNI)	Inadequate (*n* = 48)	124.5 ± 97.7	106.9 ± 82.7	8.7 ± 7.3	2.3 ± 2.8	23.1 ± 23.9
	Adequate (*n* = 12)	92.1 ± 82.9	92.7 ± 103.7	7.1 ± 7.6	1.1 ± 1.4	14.5 ± 12.9
	Excessive (*n* = 8)	83.1 ± 32.0	100.6 ± 102.4	9.7 ± 4.1	1.9 ± 2.1	19.1 ± 13.3
Phosphorus (% RNI)	Inadequate (*n* = 20)	163.5 ± 102.0^b^	143.0 ± 98.3	10.1 ± 5.7	3.5 ± 3.4^b^	26.2 ± 21.9
	Adequate (*n* = 14)	78.9 ± 97.9^a^	66.4 ± 32.2	6.6 ± 7.6	1.1 ± 1.8^a^	16.5 ± 25.7
	Excessive (*n* = 34)	99.1 ± 68.4^ab^	95.8 ± 89.7	8.4 ± 7.4	1.5 ± 1.8^a^	20.0 ± 19.1
Sodium (% AI)	Inadequate (*n* = 3)	181.7 ± 128.5	173.3 ± 141.9	8.9 ± 3.9	5.1 ± 4.8^b^	39.8 ± 35.8
	Adequate (*n* = 13)	165.8 ± 111.5	153.8 ± 91.4	10.4 ± 6.7	3.6 ± 3.5^ab^	22.6 ± 20.9
	Excessive (*n* = 52)	97.0 ± 77.2	87.1 ± 78.3	8.0 ± 7.2	1.4 ± 1.8^a^	19.6 ± 20.6
Iron (% RNI)	Inadequate (*n* = 9)	181.7 ± 111.3^b^	180.0 ± 92.7^b^	9.7 ± 4.5	3.0 ± 3.3^b^	37.1 ± 24.5^b^
	Adequate (*n* = 14)	130.4 ± 92.9^ab^	104.3 ± 86.4^a^	9.6 ± 6.5	3.1 ± 3.4^ab^	15.9 ± 13.9^a^
	Excessive (*n* = 45)	95.2 ± 79.5^a^	88.2 ± 80.6^a^	7.9 ± 7.6	1.3 ± 1.7^a^	19.5 ± 21.6^ab^
Zinc (% RNI)	Inadequate (*n* = 17)	160.3 ± 101.0^b^	148.2 ± 90.6^b^	9.5 ± 4.4	3.0 ± 2.8	28.2 ± 22.1
	Adequate (*n* = 18)	86.1 ± 76.8^a^	76.9 ± 62.6^a^	9.0 ± 9.2	2.1 ± 3.2	20.8 ± 27.0
	Excessive (*n* = 33)	105.2 ± 85.4^ab^	95.2 ± 91.5^ab^	7.7 ± 6.9	1.5 ± 1.8	17.6 ± 16.9

The percentage compared with the Dietary Reference Intakes for Koreans (DRIs) was calculated. One-way analysis of variance was used for the test of differences and Tukey’s post-hoc test. Different lower letters in the same column indicate a statistical difference among the groups, according to the one-way ANOVA and Tukey’s test (*p* < 0.05). Inadequate: <75% DRI, adequate: ≥75 and <125% DRI, and excessive: ≥125% DRI. EER, estimated energy requirement; RNI, recommended nutrient intake; AI, adequate intake

**Table 3 nutrients-13-01693-t003:** Linear relationships of taste thresholds and dietary nutrient intakes (%DRI) of elderly participants investigated by correlation and regression analyses.

Dietary Nutrients	Correlation Analysis (*r*)					Regression Analysis (*β*)				
	Sucrose RT	NaCl RT	Caffeine RT	Citric Acid RT	Umami RT	Sucrose RT	NaCl RT	Caffeine RT	Citric Acid RT	Umami RT
Energy (% EER)	−0.182	−0.174	−0.166	−0.255 *	−0.160	−0.475	−0.438	−0.033	−0.019 *	−0.099
protein (% RNI)	−0.204	−0.198	−0.096	−0.180	−0.087	−0.353	−0.331	−0.013	−0.009	−0.035
Fiber (% AI)	−0.233	−0.316 **	−0.101	−0.267 *	−0.191	−0.447	−0.585 **	−0.015	−0.014 *	−0.086
Vitamin A (% RNI)	−0.195	−0.209	−0.048	−0.156	−0.046	−0.416	−0.433	−0.008	−0.009	−0.023
Vitamin C (% RNI)	−0.232	−0.257 *	−0.009	−0.241 *	−0.176	−0.290	−0.001 *	−0.001	−0.008 *	−0.052
Thiamin (% RNI)	−0.171	−0.258 *	−0.113	−0.266 *	−0.135	−0.211	−0.308 *	−0.011	−0.009 *	−0.039
Riboflavin (% RNI)	−0.192	−0.240 *	−0.056	−0.217	−0.118	−0.325	−0.393 *	−0.007	−0.010	−0.047
Niacin (% RNI)	−0.203	−0.230	−0.094	−0.182	−0.085	−0.394	−0.432	−0.014	−0.010	−0.039
Vitamin B_6_ (% RNI)	−0.114	−0.154	0.022	−0.058	0.034	−0.093	−0.122	0.001	−0.001	0.007
Folic acid (% RNI)	−0.196	−0.225	−0.040	−0.231	−0.128	−0.275	−0.307	−0.004	−0.009	−0.043
Vitamin B_12_ (% RNI)	−0.222	−0.162	−0.007	−0.140	−0.021	−0.063	−0.045	0.000	−0.001	−0.001
Calcium (% RNI)	−0.219	−0.141	−0.027	−0.175	−0.088	−0.463	−0.288	−0.004	−0.010	−0.044
Phosphorus (% RNI)	−0.186	−0.152	−0.063	−0.171	−0.070	−0.230	−0.182	−0.006	−0.006	−0.020
Sodium (% RNI)	−0.172	−0.204	−0.041	−0.117	−0.025	−0.094	−0.107	−0.002	−0.002	−0.003
Iron (% RNI)	−0.226	−0.300 *	−0.124	−0.259 *	−0.160	−0.224	−0.288 *	−0.010	−0.007 *	−0.037
Zinc (% RNI)	−0.223	−0.206	−0.106	−0.181	−0.119	−0.267	−0.238	−0.010	−0.006	−0.034

Correlation coefficient (*r*) and regression coefficient (β, β). * *p* < 0.05, ** *p* < 0.01. Using linear model (LM) regression analysis, taste threshold values were added to the dependent variable, and nutrients were added to the independent variables (covariates).

**Table 4 nutrients-13-01693-t004:** Relationships of taste thresholds and Non-nutritional factors of elderly participants were investigated using correlation and regression analyses.

Non-Nutritional Factors	Correlation Analysis (*r*)					Regression Analysis (*β*)				
	Sucrose RT	NaCl RT	Caffeine RT	Citric Acid RT	Umami RT	Sucrose RT	NaCl RT	Caffeine RT	Citric Acid RT	Umami RT
Age ^a^	0.005	−0.002	−0.018	0.238	0.190	0.006	−0.010	−0.007	0.051	0.048
BMI ^a^	−0.097	0.084	0.323 **	0.002	−0.099	−0.026	0.050	0.176 **	0.040	−0.053
Physical activity level ^b^	−0.175	−0.089	−0.078	−0.191	−0.253 *	−0.499	−0.287	−0.138	−0.486	−0.788 *
Exercise level ^b^	0.081	0.094	0.081	0.031	0.016	0.141	0.160	0.152	0.031	0.025
Alcohol consumption ^b^	0.154	0.226	−0.014	0.163	0.017	0.403	0.530	−0.007	0.369	0.063
Smoking ^b^	−0.076	−0.060	−0.108	−0.095	−0.171	−0.307	−0.139	−0.300	−0.167	−0.506
Living situation										
Marital status ^b^	−0.107	−0.117	−0.112	0.055	−0.211	−0.258	−0.228	−0.231	0.204	−0.471
Cohabitation ^b^	0.033	0.070	0.053	−0.050	0.113	0.113	0.086	0.009	−0.116	0.130
Self-rated standard of living ^b^	−0.133	−0.193	−0.027	−0.092	−0.197	−0.204	−0.282	−0.017	−0.155	−0.314
Number of medicine intake ^b^	0.182	−0.072	0.272 *	0.087	−0.079	0.283	−0.144	0.498 *	0.104	−0.199
Social participation										
Hobby ^b^	−0.198	−0.269 *	−0.032	−0.186	−0.044	−0.194	−0.241	−0.034	−0.189	−0.037
Social gathering ^b^	−0.218	−0.257 *	−0.452 **	−0.324 **	−0.134	−0.388 *	−0.412 *	−0.688 **	−0.439 *	−0.198
Voluntary service ^b^	−0.090	−0.053	−0.129	0.057	−0.140	−0.107	−0.069	−0.152	0.077	−0.171
Education ^b^	−0.076	−0.300 *	−0.075	−0.089	−0.046	−0.074	−0.337 *	−0.076	−0.098	−0.055
Religion ^b^	−0.207	−0.227	−0.084	−0.229	−0.051	−0.229	−0.150	0.100	0.005	−0.002
QOL score ^a^	−0.415 ^**^	0.378 **	−0.351 **	−0.354 **	−0.407 **	−0.053 **	−0.057 **	−0.055 **	−0.053 **	−0.058 **

QOL: quality of life, correlation coefficient r, regression coefficient (beta, β). * *p* < 0.05, ** *p* < 0.01. ^a^ Using general linear model (GLM) regression analysis or ^b^ ordinal logistic analysis, taste threshold values were added to the dependent variable, and nutrients were added to the independent variables (covariates).

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
