# Peer review of "Taste Sensitivity of Elderly People Is Associated with Quality of Life and Inadequate Dietary Intake"

_nutrients, 2021, doi:10.3390/nu13051693_

Round 1

Reviewer 1 Report

The article has an original theme and in general is very well written and structured. However, I recommend making the following changes: At the beginning of the statistical analysis, the statistical test used to verify the normal distribution of the various variables described in the article should be identified. In figure 1, p is the value of p and not the description shown. The conclusion must be improved, since it seems to me to be poorly structured in relation to the results presented. In relation to food, how can the results of this work help nutritionists in the elaboration and eating plans.

Author Response

Dear reviewer, 

Reviewer 2 Report

The results of the article „Taste sensitivity of the elderly people is associated with quality of life and inadequate dietary intake.” are based on the study conducted on the group of 68 elderly people and 71 younger adults. Design of the study is good, statistical methods are selected  correctly, results are well presented, discussion is interesting. The References are complete and adequate, However I have some remarks and questions which I present below:

  1. Main concern: The Authors wrote that „QOL scores were investigated to determine their influence on all taste senses of the elderly individuals.” And „The QOL score showed an impact on all taste RTs tested in our study. The relationship between taste alterations and QOL has been examined in many studies, but the causal relationship has not yet been elucidated, and the participants mainly included cancer patients receiving therapy.” The conclusion that QOL has causal effect on all tastes is exaggerated. Correlation analysis (or regrression analysis) could not show causality. In other studies which Authors refer to, the participants were cancer patients and the disease often caused taste alteration and next worsening of QOL. The reverse and causal relationship postulated by Authors is not supported by their results.
  2. Under Table 2 – please correct „N lrb%)”.
  3. What does it mean Sex in Table1 – Male or Female? Why one sample proportions test was conducted for sex, whereas for other categorical factors (or maybe only for factors with at least 3 category – BMI) Chi-square test and Fisher’s exact test were applied? Four last variables in Table 1 has also exactly 2 categories (the same as Sex) and are presented in different manner (not as N and % but as n and %, where n denotes numer of persons in particular category not in a whole sample)
  4. Under Figure 3 is written that the sample of elderly participants contains exactly half of men (n = 34) and women (n = 34). But in Table 1 proportion of sex is eatimated as „68(48.9)” with explanation that numbers means „b N lrb%)”. So it is difficult to understand how the percentage „48.9” was calculated and what does it mean „lrb%)”
  5. Table 2 in amnuscript is wrongly numarated as Table 1.
  6. Explanation under Table 2 „Different letters (a,b) above the values indicate statistical significance.” is unclear. E.g. for Thiamine Inadequate/Adequate/Excessive levels and Citrin acid RT association what does it mean statistical significance (a and b) in the form: 4.9 ± 3.4b/ 1.1 ± 1.5a/1.4 ± 1.8a? Which groups differ?
  7. The Authors used Duncan's test which is very liberal in terms of Type I errors and does not protect the familywise error rate (though protecting the per-comparison alpha level). E.g. Tukey or Bonferroni tests could be used.

Author Response

Dear reviewer,

Reviewer 3 Report

The authors present a novel study assessing variations in taste perception between young and elderly individuals. Understanding variations in taste perceptions for the elderly can significantly impact dietary interventions, health outcomes and quality of life. 

The manuscript could be improved by addressing the below comments/questions:

How were taste threshold non-responders or response errors determined?

The 2.2. Participants’ baseline characteristics section of the Methods should be re-titled, this section doesn't really describe the baseline characteristics but instead describes some of the interview/surveys utilized. In addition, this section is unclear as to whether both groups asked about health-related behaviors? Please re-write so that it is clear what group received which questionnaires and surveys. 

How was the n value determined? A brief explanation of power calculation would be food. 

It would be helpful to have error bars on bar graph figures. 

Author Response

Dear reviewer,

Round 2

Reviewer 2 Report

I accept the manuscript in present form

This manuscript is a resubmission of an earlier submission. The following is a list of the peer review reports and author responses from that submission.